# Multimodality Imaging Assessment of Tetralogy of Fallot: From Diagnosis to Long-Term Follow-Up

**DOI:** 10.3390/children10111747

**Published:** 2023-10-27

**Authors:** Sara Moscatelli, Valeria Pergola, Raffaella Motta, Federico Fortuni, Nunzia Borrelli, Jolanda Sabatino, Isabella Leo, Martina Avesani, Claudia Montanaro, Elena Surkova, Massimo Mapelli, Marco Alfonso Perrone, Giovanni di Salvo

**Affiliations:** 1Centre for Inherited Cardiovascular Diseases, Great Ormond Street Hospital, London WC1N 3JH, UK; rmgpsm0@ucl.ac.uk; 2Institute of Cardiovascular Sciences, University College London, London WC1E 6BT, UK; 3Paediatric Cardiology Department, Royal Brompton and Harefield Hospitals, Guy’s and St. Thomas’ NHS Foundation Trust, London SW3 5NP, UK; 4Dipartimento di Scienze Cardio-Toraco-Vascolari e Sanità pubblica, University Hospital of Padua, 35128 Padua, Italy; valeria.pergola@aopd.veneto.it (V.P.); raffaella.motta@unipd.it (R.M.); 5Department of Cardiology, San Giovanni Battista Hospital, 06034 Foligno, Italy; f.fortuni@lumc.nl; 6Department of Cardiology, Leiden University Medical Center, 2300 Leiden, The Netherlands; 7Adult Congenital Heart Disease Unit, A.O. dei Colli, Monaldi Hospital, 80131 Naples, Italy; 8Experimental and Clinical Medicine Department, University Magna Graecia of Catanzaro, 88100 Catanzaro, Italy; sabatino@unicz.it (J.S.); i.leo@unicz.it (I.L.); 9Division of Paediatric Cardiology, Department of Women and Children’s Health, University Hospital of Padua, 35128 Padua, Italy; martina.avesani@aopd.veneto.it (M.A.); giovanni.disalvo@unipd.it (G.d.S.); 10Adult Congenital Heart Centre and National Centre for Pulmonary Hypertension, Royal Brompton Hospital, Guy’s and St. Thomas’s NHS Foundation Trust, London SW3 5NP, UK; c.montanaro@rbht.nhs.uk; 11CMR Unit, Cardiology Department, Royal Brompton and Harefield Hospitals, Guy’s and St. Thomas’ NHS Foundation Trust, London SW3 5NP, UK; 12National Heart and Lung Institute, Imperial College London, London SW3 6LY, UK; 13Department of Echocardiography, Royal Brompton and Harefield Hospitals, Guy’s and St. Thomas’ NHS Foundation Trust, London SW3 5NP, UK; e.surkova@rbht.nhs.uk; 14Centro Cardiologico Monzino, IRCCS, 20138 Milan, Italy; 15Department of Clinical Sciences and Community Health, Cardiovascular Section, University of Milan, 20122 Milan, Italy; 16Clinical Pathways and Epidemiology Unit, Bambino Gesù Children’s Hospital IRCCS, 00165 Rome, Italy; marco.perrone@uniroma2.it; 17Division of Cardiology and Cardio Lab, Department of Clinical Science and Translational Medicine, University of Rome Tor Vergata, 00133 Rome, Italy

**Keywords:** Tetralogy of Fallot, cardiovascular imaging, congenital heart disease, paediatric cardiology

## Abstract

Tetralogy of Fallot (TOF) is the most common complex congenital heart disease with long-term survivors, demanding serial monitoring of the possible complications that can be encountered from the diagnosis to long-term follow-up. Cardiovascular imaging is key in the diagnosis and serial assessment of TOF patients, guiding patients’ management and providing prognostic information. Thorough knowledge of the pathophysiology and expected sequalae in TOF, as well as the advantages and limitations of different non-invasive imaging modalities that can be used for diagnosis and follow-up, is the key to ensuring optimal management of patients with TOF. The aim of this manuscript is to provide a comprehensive overview of the role of each modality and common protocols used in clinical practice in the assessment of TOF patients.

## 1. Introduction

Tetralogy of Fallot (TOF) is the most common cyanotic congenital heart disease (CHD), with an incidence of 32.6 per 100 000 live births [1]. The classical TOF includes (i) a ventricular septal defect (VSD), (ii) an overriding aorta, (iii) the presence of right ventricular outflow obstruction (RVOTO), either at the infundibular, valvular, supravalvular level, or combined, and (iv) consequent right ventricular hypertrophy (RVH) [2]. Embryologically, these abnormalities are the result of an antero-cephalic malalignment of the developing outlet interventricular septum [3]. Other anatomical defects observed in association with the classic TOF include additional VSDs, atrial septal defect(s) (constituting the so-called “pentalogy of Fallot”), abnormalities of the tricuspid or pulmonary valves (including pulmonary atresia), a right-sided aortic arch, and an anomalous origin and course of the coronary arteries [4,5,6]. The pulmonary arteries are frequently hypoplastic or stenotic [7]. Most TOF cases are sporadic, with no specific underlying cause identified, although a genetic substrate (microdeletion of the q11 region of chromosome 22) with an association with other genetic syndromes (i.e., Di George syndrome) has been described, as well as a 3% risk of recurrency [8]. Usually, TOF patients present with various degrees of cyanosis, and the diagnosis is frequently made during foetal life or shortly after birth by echocardiography [2]. Surgical palliation involves a systemic to pulmonary arterial shunt to improve cyanosis [2]. Repair strategies aim instead at VSD closure with relief of the RVOTO with either pulmonary valve repair or valve-sparing procedures and a transannular patch [2]. Advances in palliative/reparative techniques have led to a significant improvement in the TOF patient’s survival, which is otherwise very poor. Nevertheless, patients with repaired TOF (rTOF) may develop several complications, including residual pulmonary or tricuspid regurgitation, pulmonary stenosis, RV dilatation and/or dysfunction, left ventricular systolic dysfunction, and aortic root dilatation, which may require further interventions [7]. Patients with pulmonary prosthetic valves have a higher risk of infective endocarditis [8]. Cardiovascular imaging has a pivotal role in the diagnosis and management of these patients, including initial diagnosis, risk stratification, pre-interventional planning, and long-term follow-up. The aim of this review is to provide a comprehensive overview of the role, advantages, and limitations of each imaging modality in TOF patients, from diagnosis to long-term follow-up.

## 2. Transthoracic Echocardiography in Tetralogy of Fallot

Transthoracic echocardiography (TTE) represents the cornerstone imaging technique throughout the lifetime of patients with TOF, from foetal diagnosis to assessment of late complications in adulthood. The wide availability, relatively low cost, and absence of contraindications make it an essential diagnostic tool, both pre-operatively and post-operatively [5].

The TTE data acquisition protocol should incorporate standard echocardiographic views from the subcostal, parasternal, apical, and suprasternal windows in combination with full sweeps and selected single planes.

### 2.1. Pre-Operative Evaluation

TTE allows for the detailed visualisation and measurement of cardiac structures, providing essential information on the severity of the pulmonary stenosis, size, and morphology of the ventricular septal defect (VSD), aortic override, and right ventricular outflow tract (RVOT) obstruction. Even though several anatomical types of VSD have been described, the typical VSD in TOF is an anterior malalignment type of outlet VSD, with the conal septum anteriorly diverging from the muscular septum. This single lesion is responsible for all the main components of the disease: a <50% override of the aorta over the muscular ventricular septum, muscular obstruction of the RVOT, and right ventricle hypertrophy (RVH) [6]. The analysis of the spectral Doppler and colour flow mapping may add important information. In cases of mild RVOTO, the interventricular shunt will be mostly left-to-right. As the obstruction increases, the shunt will first become bidirectional and then right-to-left, with possible hypercyanotic spells. The high parasternal view and suprasternal view allow the assessment of additional sources of blood flow, such as patent ductus arteriosus and aortopulmonary collaterals. The pulmonary valve is often thickened and dysplastic, with a hypoplastic annulus. If significant annulus hypoplasia exists, a transannular patch extension may be required at the time of surgical correction. Additionally, TTE aids in the identification of associated lesions, such as anomalous coronary arteries and additional VSD(s). These lesions significantly impact surgical planning and patient outcomes [7] and, if neglected, may impact post-op intensive care and require another surgery, if significant.

During the surgical correction of TOF, real-time transoesophageal echocardiography (TEE) plays a pivotal role in providing intraoperative guidance. TEE enables the assessment of the surgical repair, including evaluation of ventricular septal defect closure, confirmation of adequate relief of RVOTO, and assessment of residual intracardiac shunts.

### 2.2. Post-Operative and Long-Term Evaluation

Serial echocardiographic examinations help monitor surgical repair and assess residual defects, such as residual shunts, RVOTO, or pulmonary regurgitation (Figure 1).

Post-operatively, residual small shunts may often be detected along the edges of the VSD patch. These defects generally resolve following the endothelialisation of the patch. In addition, a small muscular VSD may be detected for the first time after surgery [9]. RVOTO may persist after surgery. While colour mapping aids in recognising the site of the obstruction at the sub-valvar, valvar, supra-valvar, or pulmonary branch level, continuous Doppler is particularly valuable in quantifying the severity of this residual lesion. The assessment of pulmonary regurgitation plays a crucial role in the long-term monitoring of TOF patients. Markers of severe pulmonary regurgitation include laminar and low-velocity regurgitant colour flow, diastolic flow reversal from pulmonary branches, a pressure half-time of less than 100 ms, and a pulmonary regurgitation index <0.77 (calculated using the regurgitation time as a percentage of the total diastolic time) [10,11].

TOF patients are prone to developing RV dysfunction and dilation due to chronic volume and pressure overload. Accurate assessment of RV size and function is critical, especially because, following TOF surgery, criteria for pulmonary valve replacement include the development of RV dysfunction or increasing dilatation [12]. However, the evaluation of RV by two-dimensional TTE is limited by geometric restrictions. For this reason, several parameters have been proposed to quantify systolic RV function, including the fractional area change, the tricuspid annular systolic plane excursion, the tissue Doppler’s wave, and the myocardial performance index [7].

### 2.3. Advanced Echocardiography in Tetralogy of Fallot

In recent times, there has been a growing emphasis on utilising advanced imaging methods to achieve a comprehensive and more precise assessment of cardiac function in TOF patients [13].

Three-dimensional echocardiography has emerged as a crucial tool in the long-term monitoring of patients with right ventricular outflow tract (rToF) abnormalities, and in certain instances, it has the potential to replace cardiac magnetic resonance imaging (CMR) [14]. Its primary utility lies in conducting volumetric measurements that yield dependable assessments of the ejection fraction for both the left and right ventricles [15,16], as well as the volume of all four cardiac chambers [15,16].

Moreover, strain imaging offers a more objective characterisation of myocardial function [13] and an opportunity to identify early signs of subclinical myocardial damage through the evaluation of longitudinal, radial, and circumferential strain in the left ventricle [17,18].

Recent studies have discovered relevant correlations between impaired clinical status and reduced left ventricular (LV) systolic deformation, as measured by speckle-tracking analysis, in patients with repaired TOF [19]. Furthermore, reduced LV longitudinal, radial, and circumferential strain have been identified as independent predictors of adverse clinical outcomes.

A reduction in septal strain has usually been observed, indicating that right ventricular (RV) dysfunction has a negative impact on left ventricular (LV) function, potentially through mechanical interdependence between the two ventricles [20].

In a large study involving 151 adult patients with repaired Tetralogy of Fallot (rToF), RV strains were compared to those of healthy individuals serving as controls. According to the study results, RV free-wall longitudinal strain is markedly reduced in patients who experienced either death or heart failure and is associated with adverse cardiac events, as indicated by a univariate Cox regression analysis [21].

Finally, different studies have observed that the mechanics of both the left atrium (LA) and right atrium (RA) are impaired in this population [22,23,24]. In particular, an association has been identified between abnormal left and right atrial compliance and a history of life-threatening arrhythmia in individuals with corrected TOF [23,24].

In conclusion, the use of strain parameters in the follow-up of repaired TOF may contribute to a more timely and targeted approach to the clinical management of these patients.

### 2.4. New Perspectives in the Echocardiographic Assessment in Tetralogy of Fallot

Afterload and preload can affect the estimation of LV systolic function both by using advanced and conventional echocardiographic parameters [25,26]. Among advanced echocardiographic parameters, LV global longitudinal strain (GLS) is less influenced by preload and afterload than LVEF, but is still affected by them [26]. Moreover, left or right ventricular bundle branch block, which is a very common finding in repaired TOF, can impact the echocardiographic estimation of LV systolic function and should be accounted for [27]. Regarding RV systolic function, even though RV free-wall and GLS showed less load dependency compared with conventional indices, they are still affected by pulmonary pressures and therefore RV afterload [28]. Non-invasive myocardial work (MW) indices have been proposed to overcome the influence of loading conditions on conventional and advanced echocardiographic indices of systolic function [27,29]. LV MW indices are corrected for LV afterload, which is estimated based on a reference curve derived from the non-invasive systolic blood pressure [27]. In addition, based on mitral and aortic valve events (i.e., opening and closure), the different phases of the cardiac cycle are identified by the software [27]. Based on these inputs, the software then synchronises the myocardial deformation throughout the cardiac cycle with the instantaneous LV pressure and creates a pressure-strain loop, which is used to derive the following non-invasive MW indices [27,29]: (a) constructive work, which represents myocardial lengthening during diastole and shortening during systole; (b) wasted work, which is due to myocardial shortening in diastole and lengthening in systole; (c) work efficiency, which is the ratio between constructive work and the sum of constructive and wasted work; and (d) global work index, which is the area within the pressure-strain loop and represents an index of the global performance of the LV throughout the cardiac cycle. Although the prognostic value of LV MW indices has been demonstrated in several cardiovascular diseases [30,31,32], their use has not yet been investigated in TOF and may represent an attractive area for future research. Similarly, MW indices can also be calculated for the RV, setting the cardiac cycle timings based on pulmonic and tricuspid valve opening and closure and deriving non-invasively the instantaneous RV pressure based on PASP [28]. RV myocardial work indices could theoretically overcome several limitations of conventional echocardiography for the evaluation of RV function in TOF. Although RV MW indices have shown an important correlation with invasive indices of RV systolic function and prognostic value, especially in patients with pulmonary hypertension [28,33], their role in TOF still needs to be investigated.

## 3. Cardiovascular Magnetic Resonance in Tetralogy of Fallot

### 3.1. Introduction

CMR is an advanced technique for imaging the cardiovascular system that can assess the structure, function, and blood flow without the use of radiation [34]. When it comes to TOF, CMR is especially valuable for analysing the volume and function of the RV, the pulmonary valve (PV), and the extent of myocardial fibrosis [35]. A CMR study involves capturing multiple image sequences and can be performed on individuals of any age. However, in cases of patients with claustrophobia, learning disabilities, or who are very young and struggling with holding their breath, general anaesthesia may be necessary to obtain comprehensive diagnostic information. In paediatric centres with high patient volume, the first CMR is typically conducted around 8–9 years of age, when children can undergo the examination without anaesthesia. Subsequent CMR studies are usually performed if new clinical symptoms arise or every 3–5 years, depending on the previous clinical findings [34,36,37].

### 3.2. CMR Sequences in Tetralogy of Fallot

In a CMR study for TOF, several cardiac sequences are used, and these may vary depending on the clinical question, but some common sequences include:

#### 3.2.1. Initial Localisers

The CMR routine commonly starts with single-shot sequences, either dark- or bright-blooded. In order to plan subsequent sequences and gather data regarding structures other than the heart, the axial, coronal, and sagittal planes of the thorax are imaged in dark- and bright-blood single-shot mode. These photos can be taken while holding the breath or while breathing naturally, depending on the patient’s cooperation [32,33,34,35,36].

#### 3.2.2. Balanced Steady-State Free Precession (bSSFP) Cine Sequence

The cine sequences are usually acquired during a breath-hold, particularly near the end of expiration. They are commonly acquired using the fundamental echo views, such as the four-chamber, LV three chambers, LV two chambers, short-axis (SAX), and RV two chamber long-axis views. Additionally, cine images can be planned on any useful plane, depending on the findings and clinical inquiries. These images allow for the evaluation of wall thickness and regional and global systolic function, as well as the calculation of the LV and RV end-diastolic and end-systolic volumes and the ejection fraction. End-inspiration acquisition may be investigated as an alternative to end-expiration in cases of reduced compliance. New free-breathing cines are another option that is becoming more prevalent [32,33,34,35,36].

#### 3.2.3. Phase-Contrast (PC) Flow Sequence

This sequence measures blood flow velocities and can assess the severity of any obstructions or abnormalities in blood flow. The sequences can be run in breath-hold or free-breathing [32,33,34,35,36,38].

#### 3.2.4. Four-Dimensional (4D) Flow Sequences

Four-dimensional flow MRI (magnetic resonance imaging) allows for the visualisation and quantification of blood flow in three spatial dimensions over time, hence the term “4D”. It provides detailed insights into blood flow velocities, directions, and flow patterns in the heart and the great vessels, which is especially relevant in the context of TOF [39].

#### 3.2.5. Three-Dimensional (3D) Whole Heart SSFP

This sequence gives a comprehensive evaluation of thoracic vasculature, and it offers detailed morphological information in CHD, particularly of the proximal coronary artery anatomy and the great vessel morphology and size. It is acquired through free breathing thanks to a respiratory navigator [40].

#### 3.2.6. Tissue Characterisation Sequences

These sequences have the ability to evaluate the presence of fibrosis or oedema in the heart. One of them is the late gadolinium enhancement (LGE) images obtained approximately 10–20 min after injecting a gadolinium-chelate contrast agent (GBCA). Images taken shortly after the injection, within the first 3–5 min (early gadolinium enhancement—EGE), are valuable for determining whether there is an intracavity thrombus. This is because both the myocardium and the cavity will appear enhanced due to the contrast injection, while the thrombus will appear hypoenhanced since it lacks blood supply.

The different contrast kinetics in normal and abnormal myocardium lead to myocardial enhancement as the contrast agent accumulates in the extracellular space. The distribution pattern of the contrast agent can offer valuable insights to discriminate between pathological processes like ischemic cardiac disease, previous inflammatory processes, and cardiomyopathy. Furthermore, this technique can aid in assessing the risk of arrhythmias. LGE sequences can be acquired in either breath-hold or free-breathing conditions, ideally in the same imaging planes as the cine images. This allows for direct comparison and improves interpretation analysis [41,42].

### 3.3. CMR Findings in Tetralogy of Fallot

CMR in patients with TOF should be performed in centres with a congenital team to ensure an adequate level of interpretation of the images [43]. The expected findings on post-surgical CMRs vary depending on the type of surgery performed. Indeed, TOF is a spectrum of diseases that can present with minimal antero-cephalic malalignment of the anterior ventricular septum to severe forms that overlap with the pulmonary atresia spectrum. The VSD is typically closed using a pericardial/Dacron patch, and if needed, the RVOT is enlarged. Depending on the degree of the pre-existing pulmonary stenosis (PS) or RVOTO, different techniques for RVOT augmentation may be used [44,45].

The surgeon may choose to perform an infundibulectomy with or without a commissurotomy in patients with mild RVOT constriction and an adequate pulmonary valve annulus. Resecting the muscle bundle obstructing the RVOT is required for this. The RVOT displays normal contractility on CMR pictures captured with bSSFP [44,45].

However, transannular patch repair is a different strategy when there is substantial stenosis in the RVOT or pulmonary annulus. The pulmonary annulus is opened during this surgery to reveal the RVOT’s anterior wall. The RVOT’s diameter is then increased by sewing a pericardial or Gore-Tex patch around the border of the defect. The growth of PR is one of this procedure’s main drawbacks. The anterior RVOT wall is seen to be thinned on CMR cine pictures, and PR can be visually assessed. Using PC sequences, the regurgitant fraction can be measured. On the cine pictures, the transannular patch appears as an akinetic or dyskinetic zone. Additionally, portions of the patch that are bright on post-gadolinium LGE pictures are a result of localised fibrosis [44,45,46,47].

An extracardiac conduit is inserted from the anterior RV wall/RVOT to the pulmonary arteries (PAs) in situations of pulmonary atresia or severe TOF spectrum. Usually valved, these conduits—which may be synthetic or biological—can develop pulmonary regurgitation later on or function as a tiny conduit as people age. Cine or PC sequences can be used to quantify the stenosis, blockage, or regurgitation of these conduits. However, the quantification may be incorrect due to conduit artefacts [37,44].

Patients with repaired TOF tend to require multiple PV replacements (PVR), and CMR plays a pivotal role in selecting the correct replacement timing [44,48,49].

The use of CMR in the most common sequela post-TOF repair is addressed below [50]:

Pulmonary regurgitation: PR is a common complication following the repair of TOF, and it can have negative effects on the RV, leading to RV dilatation and a negative coupling effect. Various methods are used to evaluate the severity of pulmonary insufficiency, combining information from cine and PC sequences in CMR. To assess PR, through-plane PC imaging is performed at the cross-section of the pulmonary artery just above the pulmonary valve. PR is calculated as the ratio of retrograde flow volume to antegrade flow volume. In the absence of shunts, the stroke volumes of both ventricles are equal. For a visual estimation of PR, an RVOT inplane cine can be useful together with other sequences [46,47,48].

Residual RVOT Stenosis: The most common late complication after infundibulectomy. It leads to right ventricular hypertrophy (RVH), exercise intolerance, and arrhythmias. CMR cine bSSFP sequences can identify stenosis and RVH. PC sequences can calculate velocities and gradients across the stenosis [46,47,48] (Figure 2).

RVOT Aneurysm: Severe RVOT aneurysms can lead to insufficient RV circulation. CMR cine images and post-gadolinium LGE images can visualise, respectively, the aneurysm and surrounding fibrosis [46,47,48].

Tricuspid Regurgitation (TR): TR can be associated with PR. It is important to monitor it in the presence of PR, as it can indicate the progress of PR or RV dilatation. The best way to quantify the TR is RVSV—pulmonary forward flow/100. Indeed, a visual assessment is complementary [46,47,48].

Right Ventricular Dilatation and Dysfunction: This can result from various factors such as PR, TR, RVOT aneurysm, and fibrosis of the RV-free wall. CMR cine images and volumetry assessment quantify RV dilatation and low ejection fraction [46,47,48]. RV restrictive physiology (RVRP) refers to abnormalities in RV diastolic function observed after initial repair and during late follow-up. Initially linked to end-diastolic forward flow (EDFF) into the Pas, that was notable both at CMR and echocardiography. The RVRP needs to be suspected when, despite the presence of a long-standing PR, the RV is not dilated. These are the cases where TR needs to be closely monitored [49].

Recurrent/Residual VSD: Cine pictures reveal dephasing jets over the VSD to evaluate the shunt, and CMR aids in assessing the integrity of the VSD patch. Estimating Qp:Qs (pulmonary/systemic output) can be used to determine the importance of the shunt. This is often accomplished using PC sequences to calculate the ratio of net forward flow in the pulmonary artery to net forward flow in the aorta. CMR evaluation allows for the visualisation of small VSDs [46,47,48].

Conduit Stenosis/Regurgitation, Residual Main Pulmonary Artery (MPA) Stenosis, Branch Pulmonary Artery Stenosis/Aneurysm (Figure 2): Cine images can identify MPA and PA anomalies, although a significant limitation can be represented by the presence of metallic material (stents in PAs, prosthetic valves) [46,47,48].

Left Ventricular Dysfunction: This is a predictor of poor outcomes. Cine images show global hypokinesia of the LV with a decreased LV ejection fraction. LGE can reveal the presence of a scar, which can be secondary to previous surgery (pulmonary artery vent or surgical emboli) or coronary artery disease [46,47,48].

Aorta assessment: TOF is part of cono-truncal diseases; hence, a degree of aorta dilatation is expected (Figure 2). The rate of dissection is significantly lower compared to the non-congenital population. The ideal sequence to assess the aorta is cine and 3D SFFP [50].

LGE is present in adults with repaired TOF, and it is related to adverse markers of outcome such as ventricular tachycardia, exercise intolerance, and neurohormonal activation. RV LGE has been significantly associated with arrhythmia [51,52]. Gohnim et al. build up a risk score calculator to identify patients with rTOF who are at high annual risk of death by using a weighted-risk score that integrates clinical, LGE in CMR, exercise performance, and brain natriuretic peptide (BNP) measurement [51].

### 3.4. Pitfalls in CMR

Both relative and absolute contraindications apply to CMR. For patients with ferromagnetic implants and those who have non-MRI conditional devices like pacemakers or ICDs, it is typically not advised [53]. Recent research, however, indicates that in specialised facilities with high experience, MRIs can be carried out on both conditional and non-conditional MR equipment [54]. The use of GBCA in CMR is essential for classifying myocardial sequences. Similar to other non-gadolinium-based contrast media, administering GBCA carries a minor risk of allergic responses [55]. The clinical advantage of GBCA administration should be balanced against the minor risk of developing nephrogenic systemic fibrosis (NSF), an uncommon but serious disease, in individuals with severe renal impairment (eGFR 30 mL/min/1.73 m^2^). Additionally, the buildup of gadolinium in the brain’s basal ganglia has been linked to many GBCA doses in a short period of time. The clinical importance of this discovery, however, has not yet been established [56]. Full CMR scan acquisition necessitates a high level of patient compliance, which can be difficult, particularly in the juvenile population. However, the use of free-breathing sequences and the presence of a highly skilled team enable customisation or brevity of the CMR protocol to obviate the requirement of general anaesthesia.

## 4. Cardiac Computed Tomography in Tetralogy of Fallot

Various imaging modalities have been utilised to assess TOF, including echocardiography, magnetic resonance (CMR) imaging, and cardiac computed tomography (CCT) [57,58,59]. Among these, CCT has emerged as a valuable tool for providing detailed anatomical information and functional assessment of cardiac structures [59,60].

Compared to echocardiography and CMR, CCT offers several advantages in the evaluation of TOF. Firstly, it allows for a rapid and precise visualisation of the cardiac anatomy, including the extent of the VSD, the degree of PAS, and the position of the aorta [57,59,60]. Secondly, CCT angiography enables the assessment of coronary artery anomalies, which are common in patients with TOF and play a crucial role in surgical planning [61]. Moreover, CCT provides valuable functional information, such as ventricular volumes and ejection fraction, aiding in the assessment of cardiac performance [60,61]. CCT is particularly useful for visualising the pulmonary arteries and assessing their patency and size [59], especially in the presence of stents that may hinder lumen visualisation by CMR. This information is crucial for determining the suitability of corrective surgery and identifying potential post-surgical complications.

### 4.1. Protocols

Modern, advanced scanners like the dual-source and wide-detector CT enable swift coverage of large anatomical volumes with highly detailed spatial resolution and lower temporal resolution. This results in quicker image acquisition, even in a single heartbeat, and reduces artefacts caused by heart and respiratory movements, minimising the need for ECG-gated scans, general anaesthesia, sedation, and breath-holding [59,60,61]. The typical scan range spans from the lung apices to the diaphragms, but adjustments can be made based on the area of interest, particularly for coronary assessment. Customised protocols are essential for neonates and young children with pre- or postoperative TOF due to their small size, rapid heart rate, and tendency to move during the procedure [60,62,63,64].

For detailed cardiovascular assessment and visualisation of coronary arteries, a dosage of 1–2 mL/kg of non-ionic contrast agent is commonly administered at an injection rate of 0.5 to 5 mL per second, depending on the patient’s age and size. In adults, peripheral intravenous access with an 18-gauge catheter is preferred, while in younger patients, slower injection rates are used. Power injectors are recommended for access of at least 24-gauge, and pressure-limited injection via central lines is considered safe [60,63,64]. Typical injection protocols involve biphasic injections (contrast followed by saline), while triphasic injections are used for visualising right-heart structures. To trigger the scan, bolus tracking is favoured, with manual tracking preferred over automatic tracking. Adult protocols typically use high tube potential (100 kV to 120 kV) to minimise image noise, while paediatric protocols utilise a lower dose (70–80 kV) unless the child is overweight [60,65,66]. Cardiac gating, employing prospective electrocardiogram (ECG) triggering or retrospective ECG-gating, is used to reduce motion artefacts and enable functional assessment. The latest generation of scanners allows prospective-triggered acquisitions even in paediatric patients with higher heart rates (over 100 bpm). For complex congenital heart disease cases requiring high-definition imaging, a “combo” CT protocol may be chosen, consisting of a limited ECG-triggered scan focusing on the region of interest, followed by a non-gated spiral examination of the entire thorax during the venous phase [67,68]. Scanners are equipped with various dose reduction techniques, including anatomy-based tube current modulation and iterative reconstruction algorithms [65,67,68].

Despite its advantages, exposure to ionising radiation remains a concern for paediatric patients. However, newer generation scanners significantly reduce organ doses, achieving a single-heartbeat acquisition of the whole heart volume with an effective dose estimate of less than 1 mSv [69].

### 4.2. Pre- and Post-Operative Evaluation

CCT is helpful in both pre- and postoperative assessments of TOF. CCT is invaluable in pre-surgical planning, enabling surgeons to obtain a clear roadmap of the heart’s complex anatomy before corrective surgery (Figure 3). It assists in choosing the appropriate surgical approach and minimising intraoperative risks. An innovative application of CCT in TOF management involves the integration of CT data with 3D printing technology [70]. This approach allows for the creation of patient-specific 3D models, facilitating preoperative planning and enhancing the surgeon’s understanding of complex anatomical variations.

In preoperative patients, CCT provides excellent imaging of the RVOT, the pulmonary arteries, including distal branches, the pulmonary veins, PDA, or aortopulmonary collaterals (MAPCAs), or extrinsic vascular tracheobronchial compression [59].

TOF patients may have associated coronary artery anomalies. CCT can potentially visualise the coronary arteries, aiding in the detection of abnormalities such as coronary artery anomalies or stenosis [64,65,71]. It is also ideal for assessing aortic arch and great vessel origins, like anomalous venous drainage or anomalous coronary origin, that should be known prior to surgery [68,71,72]. Evaluation of the coronary arteries is important for planning surgical intervention since an anomalous coronary origin occurs in 6–12% of TOF patients; most aberrant coronaries in TOF cross through the RVOT and may preclude surgical procedures on the RVOT. Moreover, coronary arteries should be clearly defined prior to transcatheter pulmonary valve placement due to the potential compression with device placement [60,64,71,73,74,75,76].

In a preoperative setting, CCT represents a useful tool in assessing the extracardiac structures, such as the PAs (for presence, location, and degree of stenosis), the aortic root that is relatively common, a patent ductus arteriosus, the aorto-pulmonary collaterals, or concomitant airway or lung parenchymal abnormalities [61,73].

After surgical correction, CCT can be used for postoperative follow-up to assess the success of the procedure and detect any potential complications or residual defects [64]. In postoperative patients, CCT can be used to quantify ventricular volumes, detect postsurgical complications, and plan for repeat interventions. CCT represents a practical alternative when echocardiography and CMRI are not indicated or suboptimal and is preferred for assessing graft material, calcifications, or stents in the conduit [64,65]. CCT in infants could also be used to evaluate anatomical relationships with the sternum and the chest after initial surgical reconstruction and to plan sequential stages of initial repair [61,73].

After complete repair of TOF, common postoperative complications such as RVOT dilatation, residual VSD, residual or recurrent RVOT stenosis, residual or recurrent PA stenosis, conduit stenosis, aortic dilation, stents, and implanted devices could be easily highlighted by CCT, as well as the anatomical assessment prior to transcatheter pulmonary valve replacement planning or redo sternotomy [61,64,65,66,74]. While CCT provides excellent anatomical details, it may have limitations in evaluating the dynamic changes and hemodynamic consequences of TOF. Therefore, other imaging modalities, such as echocardiography and cardiac magnetic resonance imaging (MRI), can complement these aspects.

In conclusion, cardiac CT provides detailed anatomical information, assesses coronary anomalies, and offers functional insights, making it an invaluable tool in the management of this complex congenital heart disease.

## 5. Cardiopulmonary Exercise Testing

A certain degree of exercise intolerance has been reported in most patients with rTOF, and this seems to worsen over time [77,78]. Exercise limitation in this population has a multifactorial origin: it has been related to the degree of PR [79,80,81], to biventricular dilatation and function in some studies [80,82,83,84], although these results were not confirmed in others [84]; chronotropic impairment [85] and altered lung function [86] also contribute; and parental and social barriers to physical activity may play a role in increasing physical deconditioning as well [87].

Subjective evaluation of symptoms in these patients may be inaccurate; thus, exercise tests are routinely performed to assess them objectively. Particularly, cardiopulmonary exercise testing (CPET) is currently recommended as part of the follow-up of patients with TOF to monitor patients and support treatment decisions, such as pulmonary valve replacement [88,89].

CPET assesses cardiorespiratory fitness (CRF) using different quantifiable parameters: maximum oxygen consumption (peak VO2), both as an absolute value (usually expressed in mL/kg/min) and as a percentage of its predicted value (%); ventilatory efficiency [expressed as the ventilation to carbon dioxide output (VE/VCO2) slope], ventilatory thresholds, oxygen uptake efficiency slope (OUES), and other physiological responses to an exercise stimulus, such as heart rate reserve and exercise oscillatory ventilation [90]. Reference values for peak VO2 are available both for children and adults with rTOF, and, for the latter, percentiles of expected exercise capacity have also been published [91,92,93].

A recent review, including 21 studies in adults and children with rTOF, reports a marked reduction in exercise tolerance and functional capacity, with an overall mean peak predicted VO2 of 68 ± 2.8%, which is considered a mild impairment [91]. However, this value should be interpreted cautiously because of the large heterogeneity of data available in the literature. Indeed, the use of different exercise protocols (bicycle versus treadmill) [85,93,94,95,96] and the evaluation of different parameters in various studies that include adult [94,96], paediatric [97,98], or mixed patients [85,95,99] operated with different surgical techniques [96] makes comparisons challenging. Also, most of the available studies are single-centre and retrospective.

The role of CPET in predicting major adverse cardiovascular events (MACE) in rTOF has been explored in a few studies, according to a recent meta-analysis [100]. Overall, peak VO2 or its predicted value, VE/VCO2 slope, and OUES seem to be predictive of death, event-free survival, and/or cardiac hospitalisations [95,101,102,103]. However, it is still unclear how best to integrate these exercise parameters into management algorithms to support clinical decisions, such as pulmonary valve replacement (PVR) [104]. This would be of great interest, especially in asymptomatic patients, where the identification of early sub-clinical dysfunction with objective parameters may represent a boost for PVR. Nevertheless, it should be noted that a real benefit in terms of exercise capacity and CPET results after PVR has not been demonstrated yet [105,106].

In conclusion, CPET is a valid test to follow up patients with rTOF, but standardised protocols and implementation of this technique in daily clinical practices are needed to improve risk stratification and determine outcomes.

## 6. Conclusions

Cardiovascular imaging is crucial to the diagnosis, risk stratification, and short- and long-term management of TOF patients. Several imaging modalities have an established role in this context. Echocardiography is the first-line imaging modality for diagnosis and serial rTOF evaluation, and in paediatric patients, it is most of the time the only imaging modality due to its widespread availability, low cost, and absence of radiation. Cardiovascular magnetic resonance is being increasingly used as a gold standard for volumetric assessment and shunt quantification to guide intervention and provide an assessment of structures difficult to image with echocardiography (i.e., PA branches). In addition, CMR can uniquely provide tissue characterisation, assessing the presence of myocardial fibrosis.

CCT is becoming more and more popular in the management of TOF patients for periprocedural management, in the presence of a stent, in cases of contraindication to CMR, or whenever there are doubts regarding the coronary arteries. Finally, CPET is a useful tool to guide therapeutical management and provide prognostic information. The choice of the most appropriate modality (or combination of modalities) should therefore take into account multiple factors, including the clinical question to be addressed, patients’ characteristics and contraindications, and local availability and expertise.

## Figures and Tables

**Figure 1 children-10-01747-f001:**
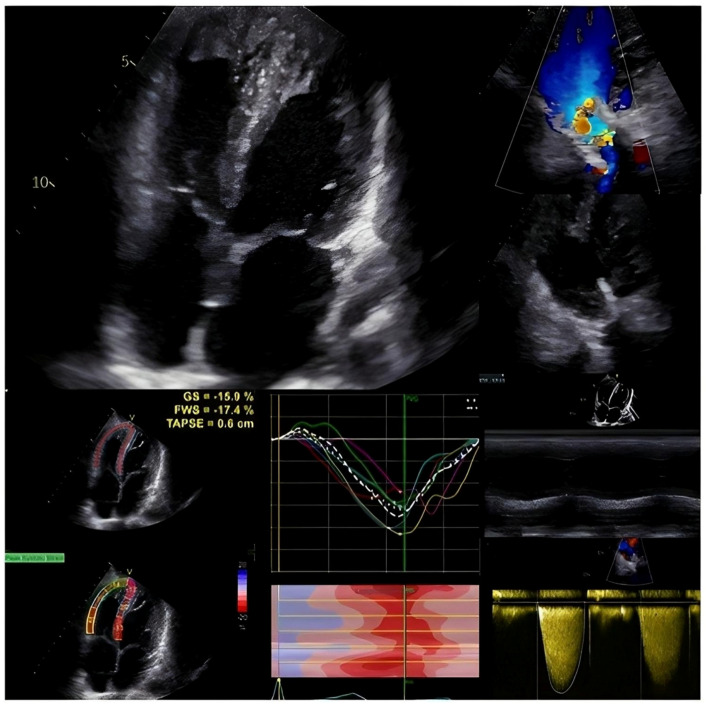
Echocardiographic assessment of a patient with TOF and previous transcatheter pulmonary valve replacement (melody): in the left upper corner a right-ventricular focused four-chamber view; in the right upper corner a colour Doppler imaging of the right ventricular outflow tract with Melody valve implantation; in the left lower corner a global longitudinal strain of the right ventricle; in the right lower corner the TAPSE evaluation and the gradient assessment through the Melody valve.

**Figure 2 children-10-01747-f002:**
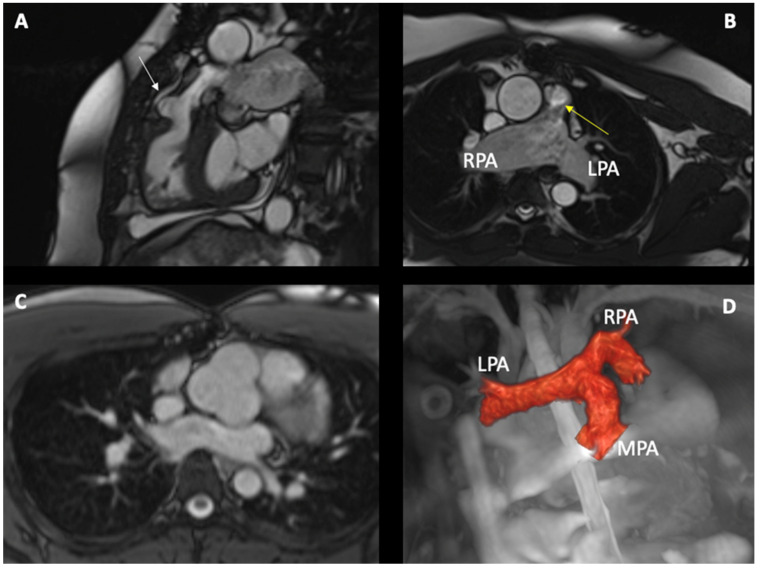
A 40-year-old patient born with TOF and an absent pulmonary valve, post complete TOF repair (7 years of age). Subsequent pulmonary valve replacement with a 23 mm aortic homograft at the age of 32 due to severe pulmonary regurgitation: (**A**) bSSFP sequence showing unobstructed RVOT with patch used during the repair (white arrow) (**B**) bSSFP cine sequence acquired in the transaxial plane illustrating dilated PA (right PA (RPA) and left PA (LPA)) and mild acceleration flow in the main pulmonary artery (yellow arrow). (**C**) bSSFP cine sequence showing a dilated aortic root. (**D**) A second patient was born with TOF and underwent total repair in the first years of life. The 3D whole heart sequence to show the PAs anatomy.

**Figure 3 children-10-01747-f003:**
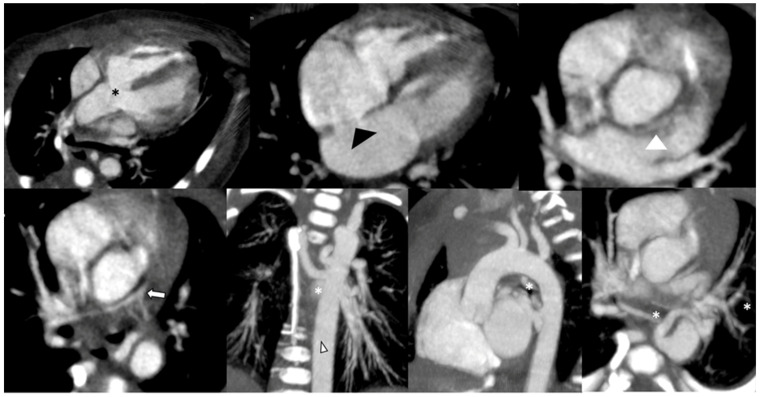
CCT in a 2-day-old newborn with Tetralogy of Fallot, scanned in sedation and free-breathing at high heart rate (145 bpm), using a low-dose protocol (80 kV) and prospective ECG triggering. The main features are also detected and depicted in the images in multiplanar and curved reformation (MPR-CPR) and maximum intensity projection (MIP): ventricular septal defect and overriding aorta (black asterisk), PFO (black arrowhead), pulmonary valve atresia and pulmonary artery hypoplasia (white arrow), and major aortopulmonary collateral arteries (white asterisks). Despite the very high heart rate, in a low-birth-weight preterm baby (2.3 Kg) scanned at a high heart rate, the left coronary artery is detectable (white arrowhead).

## Data Availability

Not applicable.

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
