# Peer review of "Multimodality Imaging Assessment of Tetralogy of Fallot: From Diagnosis to Long-Term Follow-Up"

_children, 2023, doi:10.3390/children10111747_

Round 1

Reviewer 1 Report

Comments and Suggestions for Authors

The review paper on multimodality imaging assessment of Tetralogy of Fallot: from the diagnosis to long term follow up is well written. The authors have thoroughly reviewed the role of transthoracic echocardiography, Cardiovascular magnetic resonance, cardiac computed tomography and cardiopulmonary exercise test in exceptional detail, the appropriate use of these modalities and the pros and cons of these imaging modalities. The authors could consider including the role of lung perfusion scan to assess the differential perfusion to branch pulmonary arteries before and after surgical or percutaneous balloon angioplasty.

There are a few minor errors noted.

Line 348 – “secondary to progress surgery” to be changed to “secondary to previous surgery”.

Line 348 –  (vent or surgical emboli)- term “ vent” was not clear  

Figure 2 legend– caption C is described after caption D

Line 489- pulmonary artery ipoplasia – to be edited to pulmonary artery hypoplasia

Line 491 – left coronary artery is clearly detectable – white arrowhead – it seemed not very clear in the image and may need a better image

Comments on the Quality of English Language

The quality of English was particularly good. 

Author Response

Dear Reviewer,

Thanks for your comments. We have corrected the paper accordingly. For what concern line 491, we have rephrased the sentence. Indeed, it was challenging to identify the left coronary in this patient, a neonate of 2.3Kg with high HR. 

Thanks.

Reviewer 2 Report

Comments and Suggestions for Authors

  The manuscript is clear, relevant to the field and presented in a well-structured manner. In addition to definitions and descriptions of individual methods, the Review also contains data from clinical studies conducted in children and adults with this congenital disease. It methodically meets the criteria set for this type of articles. The used images of the described imaging methods are chosen appropriately. The manuscript provides a comprehensive view of the issue, I consider it useful, it also offers a view into the future with proposals for newer methods of therapy for this disease. Based on the above facts, after a minor revision in the used literary sources, I recommend publishing the article. 105 literary sources were used in the manuscript - of which up to 20 are self-citations, which in my opinion is an inadequate number and it is necessary to reduce them and possibly replace them with literary sources from other authors.

My specific comments on the manuscript are as follows:

References - a large number of self-citations

Comments on the Quality of English Language

- there are a few typos and grammatical errors

Author Response

Dear Reviewer,

Thanks so much for your comment.

I have changed 6 citations.

Thanks

Reviewer 3 Report

Comments and Suggestions for Authors

The authors have written a review of the primary forms of cardiac imaging used in the assessment and monitoring of patients with tetralogy of Fallot. The review nicely describes the use of these modalities in patients with TOF. Some minor edits are required to improve the readability of the paper, but overall it was well written. CPET was included here - I'm not sure that it is considered an imaging modality. It is certainly important in the follow up of TOF patients. To include it, you might need to slightly alter the objective of the paper or add exercise related imaging, such as stress echo or other cardiac stress imaging. 

Specific feedback:

pg 2, line 56 - "displaced anteriorly" is redundant following "antero-cephalic malalignment"

pg 2, line 66 - "Surgical palliation requires a systemic to pulmonary arterial shunt to improve cyanosis" - requires almost implies that all patients get this procedure; should consider involves or includes?

pg 2, line 68 - should include transannular patch in the list

pg 2 line 91 - remove the comma: "size, and morphology"

pg 2 line 97 - remove a before muscular and right ventricle

pg 3, line 105 - it may be better to exclude the mention of a specific z-score here, as some surgeon's may use different z-score criteria as part of their decision to do a transannular patch

pg 3, 126 - may be helpful to define the pulmonary regurgitation index

pg 5 - it would be helpful to describe more about Figure 1 in the legend or text

pg 5 line 211 - remove "right" before RV

pg 5 212 - might be worded better as the "extent of myocardial fibrosis"

pg 5 line 220-21 - The last sentence of the paragraph is vague, but also redundant after the previous 2 sentences.

pg 6 line 240 - missing the word are before also

pg 6 line 256-57 "it offers is uniquely" needs to be edited

pg 7 lines 279-87, 290-94, 299-302 - these detailed descriptions of the surgical procedures seem a little out of place in the CMR section and might work better moved to the introduction section, particularly since you refer to the surgeries in the echo section as well.

pg 7 line 307 - a is missing from sequela

pg 7 line 325 - I think the first PR is meant to be TR

pg 8 figure 2 - not referenced in the text

pg 8 line 393 - I'm not sure how well CCT can show the degree of PS - maybe the size of the outflow/PAs?

pg 10 line 437-439 - "CCT is invaluable in pre-surgical planning..." CCT is not used pre-operatively in most centers for TOF. Perhaps this sentence is referring to TOF with pulmonary atresia and MAPCAs as shown in the Figure 3? It is more commonly used for this subgroup. This is the first mention of TOF/PA/MAPCAs, which presents some unique imaging challenges. If you are going to discuss these TOF variants - TOF/PA/MAPCAs or TOF with absent PV, I think more discussion of them is needed in the introduction and/or in the different imaging sections.

pg 10 line 451 - you can take out "coronary anatomy and any associated abnormality" since you say the same thing in the previous sentence.

pg 11 line 459 - "essential" - again I think this implies that CCT is a routine pre-op test, which it is not at many centers. This paragraph also seems very similar to the one before it

pg 11 line 473 - take out "or recurrent" before VSD

pg 12 line 497 - remove the "a" before chronotropic

pg 12 line 524 - "predict" should be predicted value

pg 12 line 536 - change "is crucial the" to "is crucial to the"

Comments on the Quality of English Language

needs some review for missing or extra articles

some awkward wording of sentences - most of the major ones were identified in my review

Author Response

Thanks so much for your comments, we have corrected oll the typographical mistakes you have identified. In addition, z-score has been removed, the pulmonary regurgitation indexed has been explained, Figure 1 text has been expanded, figure 2 has been referenced, and the type of surgeries have been also mentioned in the introduction but we have decided to leave them also in the CMR section for a better understanding of the MRI findings. In regards of CCT, PAS stenosis is measured as in CMR, we have rephrased essential and invaluable and added recent publications as proof of what was stated. Please find those below:

 - Han BK, Rigsby CK, Hlavacek A, Leipsic J, Nicol ED, Siegel MJ, Bardo D, Abbara S, Ghoshhajra B, Lesser JR, Raman S, Crean AM; Society of Cardiovascular Computed Tomography; Society of Pediatric Radiology; North American Society of Cardiac Imaging. Computed Tomography Imaging in Patients with Congenital Heart Disease Part I: Rationale and Utility. An Expert Consensus Document of the Society of Cardiovascular Computed Tomography (SCCT): Endorsed by the Society of Pediatric Radiology (SPR) and the North American Society of Cardiac Imaging (NASCI). J Cardiovasc Comput Tomogr. 2015 Nov-Dec;9(6):475-92. doi: 10.1016/j.jcct.2015.07.004. Epub 2015 Jul 23. PMID: 26272851.

- Pietro Costantini, Francesco Perone, Agnese Siani, Léon Groenhoff, Giuseppe Muscogiuri, Sandro Sironi, Paolo Marra, Serena Carriero, Anna Giulia Pavon and Marco Guglielmo. Multimodality Imaging of the Neglected Valve: Role of Echocardiography, Cardiac Magnetic Resonance and Cardiac Computed Tomography in Pulmonary Stenosis and Regurgitation. Imaging 2022, 8, 278. https://doi.org/10.3390/jimaging8100278

Reviewer 4 Report

Comments and Suggestions for Authors

This article is a REVIEW of the characteristics (findings and parameters) of the tests (ultrasound, MRI, CT) in TOF imaging studies (ultrasound, MRI, CT), from diagnosis to long-term follow-up. The main focus is not on the findings of the examinations but on the nature of the examinations. The article is very readable for beginners, but the content is also advanced and useful for pediatric cardiologists. I do not see any particular modifications.

Author Response

Thank you!
